# An Adult Mouse Model of Dilated Cardiomyopathy Caused by Inducible Cardiac-Specific *Bis* Deletion

**DOI:** 10.3390/ijms22031343

**Published:** 2021-01-29

**Authors:** Hye Hyeon Yun, Soon Young Jung, Bong Woo Park, Ji Seung Ko, Kyunghyun Yoo, Jiyoung Yeo, Hong Lim Kim, Hun Jun Park, Ho Joong Youn, Jeong Hwa Lee

**Affiliations:** 1Department of Biochemistry, College of Medicine, The Catholic University of Korea, Seoul 16591, Korea; nice1205@hanmail.net (H.H.Y.); syjjeong@hanmail.net (S.Y.J.); ted13579@hanmail.net (K.Y.); jyy7990@naver.com (J.Y.); 2Institute of Aging and Metabolic Diseases, College of Medicine, The Catholic University of Korea, Seoul 16591, Korea; 3Department of Internal Medicine, Division of Cardiology, Seoul St. Mary’s Hospital, College of Medicine, The Catholic University of Korea, Seoul 16591, Korea; lejendra@nate.com (B.W.P.); cardioman@catholic.ac.kr (H.J.P.); younhj@catholic.ac.kr (H.J.Y.); 4Laboratory Animal Research Center, Institute of Biomedical Industry, College of Medicine, The Catholic University of Korea, Seoul 16591, Korea; jsko@catholic.ac.kr; 5Department of Biomedicine & Health Sciences, Graduate School, College of Medicine, The Catholic University of Korea, Seoul 16591, Korea; 6Integrative Research Support Center, Laboratory of Electron Microscope, College of Medicine, The Catholic University of Korea, Seoul 16591, Korea; wgwkim@catholic.ac.kr

**Keywords:** BIS, dilated cardiomyopathy, tamoxifen

## Abstract

BCL-2 interacting cell death suppressor (BIS) is a multifunctional protein that has been implicated in cancer and myopathy. Various mutations of the *BIS* gene have been identified as causative of cardiac dysfunction in some dilated cardiomyopathy (DCM) patients. This was recently verified in cardiac-specific knock-out (KO) mice. In this study, we developed tamoxifen-inducible cardiomyocyte-specific BIS-KO (*Bis*-iCKO) mice to assess the role of BIS in the adult heart using the Cre-loxP strategy. The disruption of the *Bis* gene led to impaired ventricular function and subsequent heart failure due to DCM, characterized by reduced left ventricular contractility and dilatation that were observed using serial echocardiography and histology. The development of DCM was confirmed by alterations in Z-disk integrity and increased expression of several mRNAs associated with heart failure and remodeling. Furthermore, aggregation of desmin was correlated with loss of small heat shock protein in the *Bis*-iCKO mice, indicating that BIS plays an essential role in the quality control of cardiac proteins, as has been suggested in constitutive cardiac-specific KO mice. Our cardiac-specific BIS-KO mice may be a useful model for developing therapeutic interventions for DCM, especially late-onset DCM, based on the distinct phenotypes and rapid progressions.

## 1. Introduction

Bcl-2 interacting cell suppressor (BIS), also called BAG3, has been identified as an anti-apoptotic protein [1]. Numerous in vivo studies have reported the correlation of BIS expression levels with poor therapeutic outcomes in various types of cancers [2]. BIS also exerts a critical role in muscle integrity, based on the occurrence of mutations in the *BIS* gene in some fulminant myopathy (MFM) and dilated cardiomyopathy (DCM) patients [3,4]. The substitution of leucine for proline at position 209 (P209L) in exon 3 of *BIS* is the major pathogenic genotype of the *BIS* gene associated with MFM featuring progressive muscle weakness, respiratory insufficiency, and cardiac dilatation. However, genome-wide epidemiological investigations have shown that more diverse mutations, including missense, nonsense, and deletion mutations, throughout exons 2 to 4 of *BIS* are associated with a variety of cardiovascular phenotypes in familial and non-familial DCM [3,5]. Furthermore, BIS levels in the heart are significantly reduced in patients undergoing heart transplantation, suggesting that BIS has a critical function in the heart [6,7].

We previously developed *Bis*-knockout (KO) mice using the Cre-loxP system. These mice die within three weeks of birth, exhibiting metabolic derangements [8]. There is no evidence of cardiac enlargement or massive degenerative changes in the diaphragm, quadriceps, and cardiac muscles, but the ultrastructure of the muscles in *Bis*^-/-^ mice exhibits a discontinuous arrangement of myofibrils with thick and short Z bands. Recently, Fang et al. reported that the cardiac-specific deletion of *Bis* results in the impairment of cardiac contractile function as early as 10 weeks of age, leading to DCM and heart failure [9]. Furthermore, other researchers generated mice in which *Bis* was deleted from a single allele in the heart and showed that the haploinsufficiency of BIS in the heart has no effect on echocardiographic measurements of left ventricular (LV) functions until eight weeks of age [10]. However, by 10 weeks of age, a significant increase in heart size and a decrease in the LV ejection fraction are observed, as is a decrease in the sensitivity of myocytes to an adrenergic stimulus. These homozygous and heterozygous cardiac-restricted *Bis*-KO mice decisively provided evidence of the cardioprotective function of BIS, which was unclear in the whole-body *Bis*-KO mice due to early lethality within three weeks.

However, the slow, progressive loss of cardiac function over several months in constitutive and cardiac-restricted *Bis*-KO mice may be a limitation in using them as a DCM model to determine the effectiveness of any novel therapeutic approach. Furthermore, the consequential cardiac dysfunction impairs fertility, presenting a difficulty in obtaining a sufficient number of offspring with the same genotypes. Therefore, to overcome these limitations, we aimed to generate inducible cardiac-specific *Bis*-KO mice using the α-myosin heavy chain (α-MHC)-MerCreMer system [11].

## 2. Results

We generated inducible and cardiac-specific *Bis*-KO (*Bis*-iCKO) double transgenic (*Bis^f/f^*: α-MHC-MerCreMer-positive) mice that were healthy, displaying no sign of cardiac dysfunction. Inducible deletion of cardiac *Bis* was achieved by activating cardiomyocyte-specific Cre recombinase with tamoxifen injections (Figure 1A). The depletion of BIS in *Bis*-iCKO mice was observed only in cardiomyocytes and not in other organs, such as the skeletal muscle, liver, and lung, upon tamoxifen injection (Figure 1B). While a dramatic decrease in BIS levels was observed in *Bis*-iCKO mice, tamoxifen did not appear to affect BIS levels in the cardiomyocytes or skeletal muscles in the control mice (*Bis^f/f^* mice lacking the Cre recombinase), indicating the specificity of the α-MHC-MerCreMer system. Sixty percent of *Bis*-iCKO mice survived after six doses of tamoxifen, while no deaths were observed in control mice given the same treatment (Figure 1C). The hearts from surviving *Bis*-iCKO mice were notably dilated (Figure 1D), and the ratio of heart weight to body weight was significantly increased in *Bis*-iCKO mice compared with that in control mice (Figure 1E). H&E staining also revealed marked cardiac enlargement and decreased thickness of the anteria, posteria, and septal walls in *Bis*-iCKO mice (Figure 1F).

The cardiac function of *Bis*-iCKO and control mice was evaluated by serial echocardiography (Figure 2A). Before tamoxifen treatment, there was no difference between the control and *Bis*-iCKO mice. However, the LV systolic function was progressively decreased in *Bis*-iCKO mice after tamoxifen treatment. On day 28, after six doses of tamoxifen, the LV ejection fraction (LVEF) and fractional shortening (FS) decreased from 86% to 36% and from 50% to 14%, respectively, in *Bis*-iCKO mice, while no decrease was observed in control mice (Figure 2B,C). Concomitant with serious LV dysfunction, the LV internal dimensions during systole and diastole were significantly increased in *Bis*-iCKO mice compared with those in control mice (Figure 2D,E), which is consistent with the H&E staining result (Figure 1F). The time-dependent reduction in cardiac function in iCKO mice was correlated with the decline in BIS levels, as shown by Western blotting assay and qRT–PCR (Figure 2F). Collectively, our results indicated that the inducible deletion of cardiac-restricted *Bis* resulted in the development of DCM in adult mice, represented by a massive decrease in LV function and an increase in adverse remodeling.

We examined the ultrastructure of the myocardium from *Bis*-iCKO mice by electron microscopy (Figure 3A). The myofibril appeared distorted, with a lower density, and the integrity of the Z-disk was disrupted by wider Z-lines. In addition, mitochondria were located in the spaces between myofibrils in a dispersed pattern in *Bis*-iCKO cells, unlike the compact array observed in control, indicating cardiac dilatation. The quantitative RT-mRNA assay showed that the mRNA levels of atrial natriuretic factor and B-type natriuretic peptide, two markers for heart failure, were notably increased in *Bis*-iCKO mice at 4.1 and 3.7 times larger than those of the control, respectively (Figure 3B). The mRNA levels of the stress–response gene β-MHC, as well as the profibrotic genes collagen α1 type I (Col1a1) and III (Col3a1), were also significantly higher in the myocardium of *Bis*-iCKO mice than in the myocardium of the control. The alteration of these mRNA represented the progression of cardiac remodeling after the onset of DCM.

Previously, the constitutive deletion of *Bis* in the heart was shown to accelerate the degradation of small heat shock protein (HSP) levels, which is essential for the chaperone complex, leading to the aggregation of proteins that are critical for cardiac function, such as LDB3, ENH, α-Actinin, MHC, DESMIN, VINCULIN and CAPZβ [9]. We also examined the expression profiles of HSPs and found that the decrease in the levels of small HSPs such as HSPB8 ((but not HSP70) paralleled the accumulation of DESMIN, a major intermediate filament essential for myocardial contractility [12], in the insoluble fraction in the *Bis*-iCKO mice (Figure 4). Thus, as is the case with the constitutive cardiac restriction of BIS, the disturbance in the cardiomyocyte protein homeostasis might be the molecular basis by which BIS loss leads to cardiac dysfunction. This aligns with the findings of a previous study in skeletal muscle cells showing that BIS is required for the preservation of the Z-disk structure via the chaperone-assisted degradation of Z-disk proteins such as filamin [13].

Given that cardiac dilation can be caused by Cre expression alone as an off-target effect [14], we validated cardiac function in α-MHC-MerCreMer transgenic mice following tamoxifen injections. As shown in Appendix A, we did not find any evidence indicating cardiac dysfunction in the α-MHC-MerCreMer transgenic mice, based on the echocardiography results as well as several cardiac gene expression profiles after treatment with six doses of tamoxifen. Thus, the DCM induction observed in the *Bis*-iCKO mice may be a direct outcome of reduced BIS expression, independent of the toxic effect of Cre expression.

## 3. Discussion

Our results reinforced the critical role of BIS in cardiac function indicated by its frequent mutations in DCM patients and reduced levels in failing hearts [5,6,7]. In addition, our research also provided evidence of the stabilizing effect of BIS on sHSPs, which is an essential element in maintaining the protein quality control (PQC) of cardiomyocytes, as demonstrated previously [9]. In terms of the two main phenotypes featuring DCM development accompanied by disturbances in PQC, either constitutive or inducible heart-specific *Bis*-KO mice may be useful in vivo mouse models for studying DCM pathology, BIS, and cardiac function.

Our system may have several advantages over the constitutive cardiac deletion of *Bis* as a DCM model. First, our model may be more appropriate for late (adult) onset DCM research because the tamoxifen injection starting time was temporally controllable. Second, the tamoxifen injection protocol was quite simple and was adjustable according to the resulting cardiac phenotypes. Even though six tamoxifen injections were scheduled to induce maximum Cre recombinase activation, our echocardiographic data clearly showed that the LV function and remodeling significantly deteriorated at days 14 and 21, before the fifth tamoxifen injection (Figure 2). Consistently, hearts harvested from mice that died before the fifth tamoxifen injection were quite enlarged relative to those in the control group, which may suggest that the cause of death was due to cardiac dysfunction rather than tamoxifen toxicity. This was supported by the 100% survival rate of control mice. Thus, only four doses of tamoxifen were required to induce DCM within three weeks; this dosing schedule and disease progression time are probably beneficial for evaluating the outcome of therapeutic interventions for DCM. Finally, in the constitutive deletion mutants, the expression of BIS in the heart was determined as either absent (in homozygous KO mice) or half of that of wild-type (in heterozygous KO mice) at birth, indicating that the severity of DCM could not be modulated. In comparison, our system may confer flexibility to the suppression of BIS expression by enabling modulation of the tamoxifen dose and injection schedule.

Considering that women with DCM have a better prognosis than men with DCM [15], it is important to develop a model that reproduces sex-related differences in the prevalence and severity of the disease. However, a significant limitation of our study was the insufficient number of subjects, which prevented us from drawing any conclusions regarding sex-related differences. In addition, to establish an accurate DCM model that represents the different levels of severity, it is necessary to conduct a long-term follow-up of *Bis*-iCKO mice with various BIS expression levels induced by fine-tuning the tamoxifen dose and injection schedule.

As far as we know, this is the first approach to establish inducible *Bis*-KO in a tissue-specific manner. This strategy of tissue-specific *Bis* deletion may be applicable to other organs, providing a useful in vivo tool for studying the essential functions of BIS in corresponding human tissues under physiological and pathological conditions.

## 4. Materials and Methods

### 4.1. Animals

Mice homozygous for exon 3 floxed *Bis* (*Bis^f/f,^*) [16] were crossed with α-MHC-MerCreMer transgenic mice (Jackson Laboratory, Farmington, CT, USA) to generate tamoxifen-inducible cardiac-specific *Bis*-KO mice (*Bis^f^*^/f^: α-MHC-MerCreMer-positive). To achieve cardiomyocyte-specific excision of the *Bis* gene in adult mice, mice of 16–20 weeks of age received tamoxifen (20 mg/kg, dissolved in corn oil) by intraperitoneal injection in three sets of two consecutive injections, with one-week and two-week intervals between the first and second sets and between the second and third sets, respectively (Figure 1A). Sex-matched *Bis^f/f^* mice were administered tamoxifen as a control. The day of the first injection was recorded as day 1. All procedures and provisions for animal care were approved by the Institutional Animal Care and Use Committee at the College of Medicine of The Catholic University of Korea (CUMS-2017-0320-10).

### 4.2. Echocardiography

The assessment of cardiac function was performed using a transthoracic echocardiography system (Affniti 50G, Philips, Andover, MA, USA) equipped with a 15 MHz L15-7io linear transducer (Philips) under 2% isoflurane anesthesia [17]. Serial echocardiograms were performed just before (day 0) and after (days 7, 14, 21, and 28) tamoxifen injection. LV parameters were measured using M-mode tracing.

### 4.3. Histology and Electron Microscopy

Each heart was excised, fixed in 3.7% formalin, and embedded in paraffin. From the three transverse sections of the apex, middle ring, and base, 5 μm sections were cut and stained with hematoxylin and eosin (H&E) and then mounted and imaged using the laser scanning microscope LSM 880 NLO with Airyscan (Zeiss, Jena, Germany). Ultramicroscopic observation was performed using a transmission electron microscope (TEM; JEM 1010, Tokyo, Japan) operating at 60 kV, and images from the TEM were recorded with a CCD camera (SC1000, Gatan, Pleasanton, CA, USA).

### 4.4. Western Blotting Assay

Immunoblotting and fractionation were performed according to the standard procedure [18]. Briefly, mouse tissues were lysed with lysis buffer, homogenized by sonication, and then subjected to SDS–PAGE. The primary antibodies used in this study are listed in Appendix A. The immunoreactive bands were visualized by enhanced chemiluminescence following exposure to the peroxidase-conjugated secondary antibody using an automatic film processor (Pro-MG, Desung Tech, Sungnam, Korea).

### 4.5. Quantitative Real-Time PCR (qRT–PCR)

Total RNA was isolated from the LV of mouse hearts and processed for cDNA synthesis using the PrimeScript RT master mix (Takara-Bio, Kyoto, Japan). qRT–PCR was performed using SYBR green premix Ex Taq (Takara-Bio) with specific primers (Appendix A) on the ABI Prism 7300 sequence detection system (Applied Biosystems, Carlsbad, CA, USA). The expression level of each mRNA was determined using the ^ΔΔ^Ct method after normalization to the Ct value for 18S rRNA.

### 4.6. Statistics

Data are expressed as the mean ± standard deviation (SD). Student’s *t*-test was used to compare data between the two groups. A *p*-value of less than 0.05 was considered statistically significant.

## 5. Conclusions

We established cardiac-specific and inducible *Bis*-KO mice that exhibited significantly impaired LV contractility and adverse remodeling, providing further evidence of the critical role of BIS in cardiac function. Our temporally controllable KO mice may serve as a new inducible model for studying DCM pathology.

## Figures and Tables

**Figure 1 ijms-22-01343-f001:**
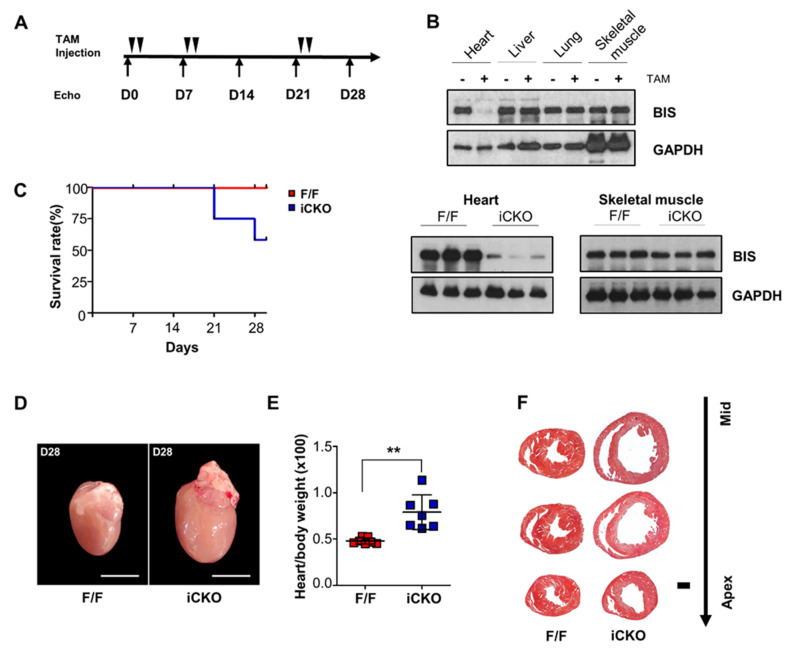
Cardiac specific depletion of BCL-2 interacting cell death suppressor (BIS) by α-MHC-MerCreMer strategy. (**A**) Scheme of tamoxifen (TAM) injection and echocardiography (Echo). (**B**) Western blot analysis of BIS expression in several tissues from double transgenic mice (Bis^f/f^: α-MHC-MerCreMer-positive) before and 28 days after tamoxifen injection (upper) and in the hearts and skeletal muscles from control (Bis^f/f^, F/F) and tamoxifen-inducible cardiomyocyte-specific BIS-KO (Bis-iCKO) (iCKO) mice 28 days after the first tamoxifen injection (lower). (**C**) Survival curves for F/F (red) and iCKO (blue) mice. Survival was analyzed every week after tamoxifen injection (n = 12 for both groups 6 males and 6 females). (**D**) Morphology of whole hearts from F/F and iCKO mice on day 28; scale bar, 2 mm. (**E**) Heart weight to body weight ratio for F/F and iCKO mice was determined 28 days after the first tamoxifen injection (n = 7 for both groups, 4 males and 3 females). Data are presented as the mean ± standard deviation (SD). ** *p* < 0.01. (**F**) Hematoxylin and eosin staining of serial sections of the hearts from F/F and iCKO mice 28 days after tamoxifen injections; scale bar, 1 mm.

**Figure 2 ijms-22-01343-f002:**
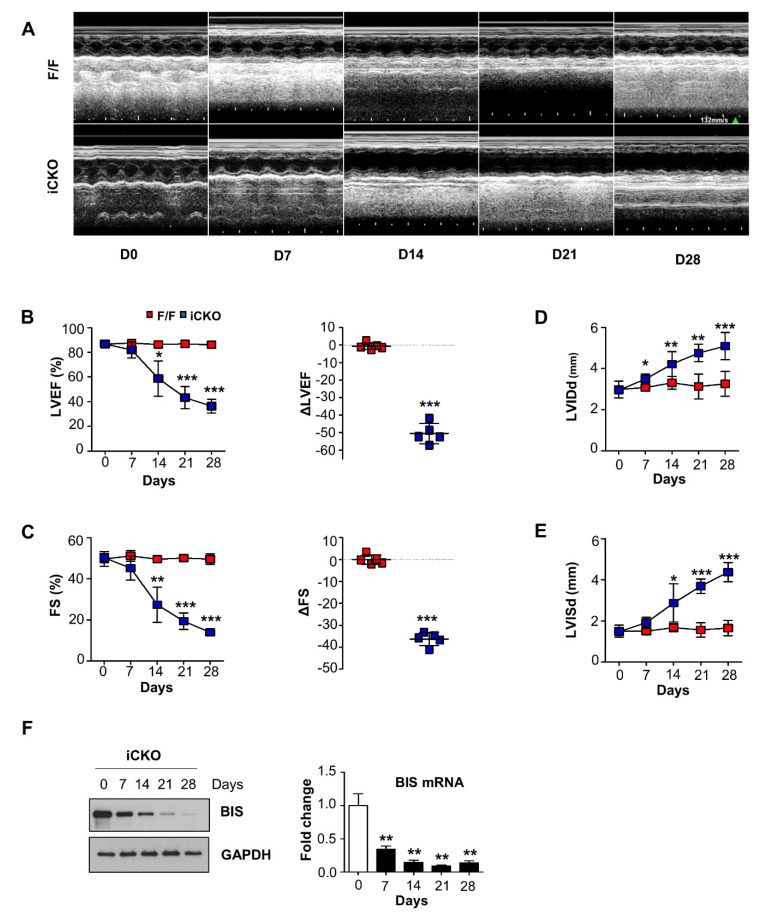
Tamoxifen-inducible depletion of BIS in the cardiomyocytes resulted in the development of dilated cardiomyopathy (DCM) in adult mice. (**A**) Echocardiographic tracing of F/F and iCKO mice before (D0) and after tamoxifen injection (D7, D14, D21 and D28). (**B**,**C**) Values of left ventricular ejection fraction (LVEF) and fractional shortening (FS) were obtained using echocardiography (left panels; n = 5 for both groups, 3 males and 2 females). LVEF delta and FS delta changes on day 28 are shown in the right panels. (**D**,**E**) Left ventricular internal diastolic dimension (LVIDd) and left ventricular internal systolic dimension (LVISd) were progressively increased in iCKO mice. Data are presented as the mean ± standard deviation (SD). * *p* < 0.05, ** *p* < 0.01, and *** *p* < 0.001 compared with the values from F/F mice on corresponding days. (**F**) Time-dependent BIS expression levels in iCKO mice after tamoxifen injection were determined using Western blotting assay (left) and qRT–PCR (right). ** *p* < 0.01 compared with the values on D0.

**Figure 3 ijms-22-01343-f003:**
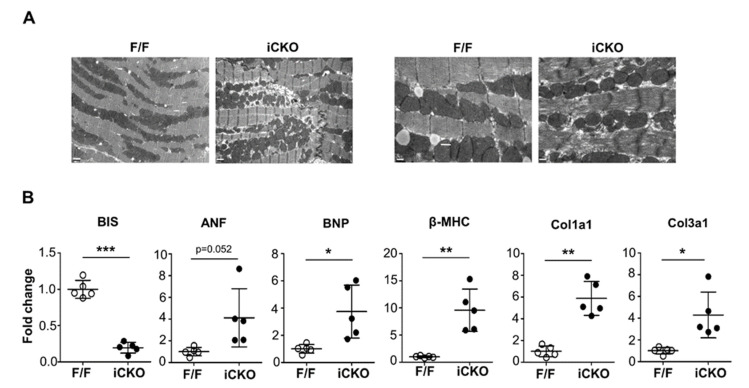
Ultrastructure analysis and expression profiles of cardiac mRNA in *Bis*-iCKO mice. (**A**) Representative transmission electron microscopy images of the myocardium of F/F and iCKO mice 28 days after tamoxifen injection. Scale bar, 1.0 μm (left) and 0.5 μm (right). (**B**) Quantitative analysis of the mRNA levels of cardiac failure markers and profibrotic genes in F/F and iCKO mice 28 days after tamoxifen injection (n = 5 for both groups). Data are presented as the mean ± standard deviation (SD). * *p* < 0.05, ** *p* < 0.01, and *** *p* < 0.001.

**Figure 4 ijms-22-01343-f004:**
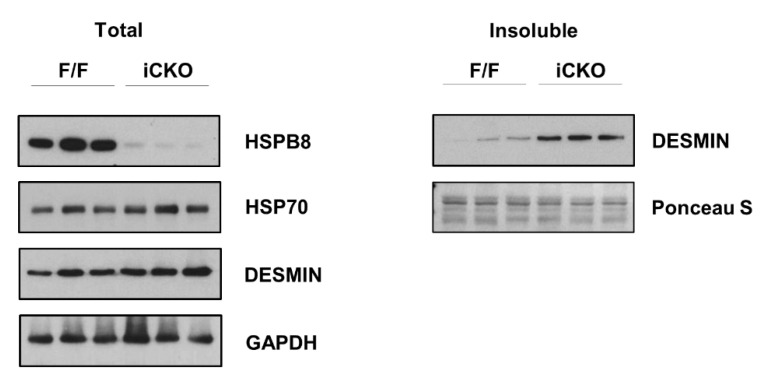
Decreased expression of HSPB8 and subsequent accumulation of insoluble DESMIN in the cardiomyocytes in *Bis*-iCKO mice. Immunoblots for HSPB8, HSP70, and DESMIN in total fractions from the hearts of F/F and iCKO mice (**left**). Increased levels of DESMIN in the insoluble fraction in the hearts of iCKO mice were demonstrated using Western blotting (**right**).

## Data Availability

The data presented in this study are available on the request from the corresponding author.

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
