# Peer review of "An Adult Mouse Model of Dilated Cardiomyopathy Caused by Inducible Cardiac-Specific Bis Deletion"

_ijms, 2021, doi:10.3390/ijms22031343_

Round 1
Reviewer 1 Report
Hye Hyeon Yun et al. An adult mouse model of dilated cardiomyopathy caused by inducible cardiac-specific Bis deletion
Hye Hyeon Yun et al generated a tamoxifen inducible and cardiac-specific Bis-KO (Bis-iCKO) double transgenic (Bisf/f: α-MHC-MerCreMer-positive) that resulted in the development of DCM in adult mice.
The article is well conceived, well written, and the experimental design is correct. The studies carried out: Echocardiography, Histology and electron microscopy, Western blotting and Quantitative real-time PCR (qRT-PCR) clearly establish the effect of the delection and the figures are very illustrative. The paper is especially relevant in view of the recent description of mutations affecting this gene and causing dilated cardiomyopathy.
I would just ask for a little clarification. The authors claim that the constitutive deletion of Bis in the heart was shown to accelerate the degradation of small heat shock protein (HSP) levels, which is essential for the chaperone complex, leading to the aggregation of proteins that are critical for cardiac function. And they provide data related to the accumulation of desmin. I wonder if they have only focused on this protein or if it is the only one, of several studied, in which they found aggregation.
Author Response
Please see the attachement

Reviewer 2 Report
In their manuscript, Yun and co-workers generate a mouse model of dilated cardiomyopathy by an inducible deletion of Bis in the heart. After the generation and characterization of the model, the authors demonstrated that the depletion of Bis causes structural and functional alterations of the heart. Moreover, the lack of Bis modify the myocardium structure with an altered expression of gene involved in heart failure, stress-response and fibrosis, and also downregulate the HSPB8. The work is well designed and the conclusions are sustained by the obtained results. Despite a heart-specific Bis-KO model has already been generated by others, this inducible model will be useful in the study of the development of DCM, allowing to better modulate in time the expression of Bis.
A complete list of minor criticisms is detailed below.
-How were the Bisf/f and α-MHC-MerCreMe obtained? Are they commercial or were generated by the authors? Authors should report the origin of the mouse models.
-The authors reported that the ratio of heart weight to body weight was significantly increased. What about the body weight? Are there any significant differences between control and Bis-iCKO?
-There are significant alterations in cardiac function during the treatment with tamoxifen as a consequence of the Bisdownregulation. The authors showed representative echocardiographic tracing at D0 and D28. It will be interestingly to see also the image at the other time points to follow the progressive alteration of heart function. Authors should show representative pictures for each time point, if they prefer as supplementary materials.
-In the previous mouse model with constitutive Bis deletion by Fang et al. the depletion of Bis causes the upregulation of Hsp70 while in the present model it was not observed. Could authors comment this aspect?
